# In Translation: FcRn across the Therapeutic Spectrum

**DOI:** 10.3390/ijms22063048

**Published:** 2021-03-17

**Authors:** Timothy Qi, Yanguang Cao

**Affiliations:** Division of Pharmacotherapy and Experimental Therapeutics, UNC Eshelman School of Pharmacy, Chapel Hill, NC 27599, USA; timothy_qi@unc.edu

**Keywords:** neonatal Fc receptor, monoclonal antibody, pharmacokinetics, immunoglobulin G, Fc-fusion protein, immune complex, physiologically-based pharmacokinetic modeling

## Abstract

As an essential modulator of IgG disposition, the neonatal Fc receptor (FcRn) governs the pharmacokinetics and functions many therapeutic modalities. In this review, we thoroughly reexamine the hitherto elucidated biological and thermodynamic properties of FcRn to provide context for our assessment of more recent advances, which covers antigen-binding fragment (Fab) determinants of FcRn affinity, transgenic preclinical models, and FcRn targeting as an immune-complex (IC)-clearing strategy. We further comment on therapeutic antibodies authorized for treating SARS-CoV-2 (bamlanivimab, casirivimab, and imdevimab) and evaluate their potential to saturate FcRn-mediated recycling. Finally, we discuss modeling and simulation studies that probe the quantitative relationship between in vivo IgG persistence and in vitro FcRn binding, emphasizing the importance of endosomal transit parameters.

## 1. Introduction

As the most abundant serum immunoglobulin, immunoglobulin G (IgG) represents 10–20% of total plasma protein. The neonatal Fc receptor (FcRn) is broadly expressed and plays an important role in maintaining homeostatic levels of circulating IgG [1,2,3]. The Fc region of IgG binds FcRn in a pH-dependent manner, with high affinity at endosomal pH levels (~6.0) and little to no affinity at physiologic pH levels (~7.4). The pH-dependent nature of this interaction underpins a process known as FcRn-mediated recycling, which rescues endosomal IgG from degradation through binding in the endosomal lumen and sequential release in the extracellular environment. Relative to other proteins in circulation, FcRn-mediated recycling confers IgG an unusually long half-life of ~21 days. In the case of autoreactive IgG, this contributes toward the pathophysiology of autoimmune diseases [3]. Other molecules that contain an Fc domain, such as antibody therapeutics and Fc-fusion proteins, are similarly subject to FcRn-mediated recycling. Accordingly, understanding how FcRn traffics its substrates across diverse contexts is critical to measuring, interpreting, and forecasting the disposition of endogenous IgG and Fc-bearing therapeutics.

FcRn-mediated recycling begins with IgG pinocytosis; target-mediated endocytosis is considered a significantly lesser source of IgG uptake (Figure 1) [4]. Once early endosomes become acidified, the affinity between FcRn and IgG Fc increases, enabling saturable binding. Residual unbound IgG is delivered to lysosomes for degradation [5,6]. The membrane-bound IgG-FcRn complexes are, however, sorted into tubulovesicular transport carriers (TCs), which can then follow one of several pathways. Some TCs participate in the interendosomal transfer of contents, while others undergo a process known as futile looping, wherein TCs leave and return to the same endosome [7]. Other TCs are trafficked to the plasma membrane and undergo exocytosis, releasing their soluble contents into the extracellular space and exposing the FcRn-IgG complex to physiologic pH [8]. This reduces the affinity of FcRn for IgG Fc, freeing IgG and returning it to the extracellular environment. Importantly, the polarity of IgG exocytosis leads to diverse functions across tissue contexts. In any case, FcRn is rapidly retrieved from the cell surface after it releases IgG, resulting in a relatively low membrane-localized pool during steady-state conditions [9]. The majority of IgG follows this pinocytosis-exocytosis pathway [10].

## 2. FcRn Structure, Binding, and Stoichiometry

FcRn is related to major histocompatibility (MHC) class I molecules and consists of a soluble light chain β_2_ microglobulin (β2M) noncovalently bound to a membrane-bound heavy chain [11,12,13]. While 4–7% of FcRn localizes to the plasma membrane, the majority resides within endosomes, where it binds IgG Fc at its C_H2_-C_H3_ hinge in a site distinct from those where FcγR and C1q complement interact [13,14,15,16]. In the endosome, two FcRn bind a single IgG with equal affinity [12,17]. The resulting avidity contributes to the ability of FcRn to extend IgG’s half-life [17]. Conversely, FcRn deficiency in settings such as familial hypercatabolic hypoproteinemia can substantially increase the rate of circulating IgG catabolism, resulting in 31–36% daily turnover and lower homeostatic serum concentrations [18,19]. FcRn also binds albumin and, depending on its configuration, can bind both IgG and albumin simultaneously. For example, a reclined configuration has been proposed to accommodate tripartite binding to albumin and IgG, with additional mitigation of steric hindrance conferred by the flexibility of IgG’s Fab regions [20,21]. Notably, the binding of FcRn to albumin can sterically occlude and concomitantly alter the pharmacokinetic properties of highly protein-bound drugs and albumin-fused therapeutics. These interactions have been reviewed extensively [22].

### 2.1. pH Sensitivity of Binding

The pH-sensitive nature of FcRn-IgG binding is largely attributed to the titration of IgG residues His310 and His435 in the acidic environments of endosomes [23]. In physiologic pH ranges, FcRn has little to no affinity for IgG and albumin, except for mouse IgG2b and a few human IgG3 allotypes [24]. This differential affinity of human IgG3 affects its half-life and is thought to result from the substitution of His435 with Arg435, which does not deprotonate at pH 7.4 [25]. At pH 6.0, various species-specific anionic residues of the FcRn α_2_-helix, including Glu117, Glu132, Glu135, and Asp137, can form salt bridges with the newly protonated histidine residues on IgG [26]. As a result, IgG taken up by fluid-phase pinocytosis binds endosomal FcRn with nanomolar affinity [16]. Other IgG residues that can influence FcRn binding include Met252, Met358, Met428, and Ile253 [27,28,29,30,31]. Met252, Met358, and Met428, present on human IgG1 Fc, can be oxidized during the therapeutic antibody manufacturing process, potentially impairing FcRn binding [27,28,29,30]. Simultaneous mutation of Ile253, His310, and His345 (collectively known as IHH) prevents FcRn from binding at pH 6.0 and increases IgG clearance [31]. Conversely, Fc variants with increased affinity for FcRn at pH 6.0, such as YTE (M252Y/S254T/T256E), have been shown to extend IgG half-life in numerous preclinical species as well as humans [4,32]. The pH-dependent IgG/FcRn interaction has been broadly exploited to engineer therapeutic IgGs and Fc-fused proteins with extended serum persistence, which will be thoroughly reviewed in the following sections.

### 2.2. Fab Contribution to Binding

The variable region Fab can also affect binding between IgG and FcRn, although the mechanisms by which this is accomplished are still emerging [33,34,35,36,37,38]. This possibility was first raised by investigators who demonstrated that antibodies with identical Fc but varied Fab regions had different capacities to bind FcRn [33]. There is evidence supporting a role for positive charge patches, variability in isoelectric point (pI), κ vesus λ light chains, and direct Fab-FcRn interaction in mediating this differential affinity. However, the significance of these effects varies among studies [17,34,36,37,38,39]. In particular, a recent study found no relationship between positive charge distributions in the variable region and FcRn affinity in eight IgG1 antibodies. The investigators also found that the variable region of IgG1 could allosterically modulate FcRn binding by conferring differential sensitivity of *K*_a_, *K*_d_, and *K*_D_ to shifts in pH, antigen (Ag)-binding status, and the balance between enthalpic and entropic forces [38]. Curiously, the authors observed lower endosomal FcRn association rates for antibody-antigen immune complexes (ICs), which they believe serves to reduce the IC rescue rate from degradation. This purported effect of Ag binding on FcRn affinity is corroborated by another recent study that found Ag binding capable of inducing structural changes in the IgG FcRn binding region [40]. Single amino acid additions to the N- or C-termini of Ags were capable of increasing FcRn affinity by 1.1–4.6-fold over Ag-free IgG, highlighting the importance of Ag physicochemical properties on FcRn affinity. Based on these data, differential FcRn binding upon complexation with Ag may need to be considered when evaluating the disposition of antibody therapeutics.

### 2.3. Binding Modulation by Other IgG Regions 

Beyond the Fab region, other variations distal to the C_H2_–C_H3_ juncture also affect binding between IgG and FcRn [41,42]. For example, monoclonal antibody (mAb) therapeutics with G1m17,1 allotypes have been shown to more efficiently bind and be transcytosed by FcRn than G1m3,-1 allotype mAbs. These mutations in the CH1 and CH3 regions of IgG1 and are notable in that variations in either region alone did not affect FcRn binding [41]. Even mutations in the hinge connecting Fc and Fab regions have been found to modulate FcRn binding, although somewhat unpredictably and in a highly heterogenous manner [42]. Taken holistically, it would appear that an IgG molecule’s FcRn binding affinity is highly sensitive to variations along the majority of its length.

### 2.4. Species Variation of Binding

Binding between IgG and FcRn also varies across species. Human IgG1 has greater affinity for cynomolgus monkey, rat, and mouse FcRn than for human FcRn in endosomal pH ranges [17,43,44]. This divergence is believed to arise from variation in FcRn residue 137 and particularly complicates the use of mice as translational models, which exhibit 10 to 15-fold higher affinity for human IgG1 [4,17,45]. Similar behavior has also been demonstrated in IgG2 allotypes [44,45,46,47]. As a result, straightforward extrapolation of pharmacokinetic properties from data obtained in mice can be challenging unless comparable binding to mouse and human FcRn is demonstrated. These divergent binding kinetics and their subsequent effects on IgG disposition have prompted the development of transgenic mouse models that express human FcRn, which are discussed at length later in this review.

## 3. FcRn Expression and Its Effects on Homeostatic IgG Distribution

Despite its titular role in the prenatal transport of antibodies from mother to child, FcRn is widely expressed and serves diverse functions in IgG protection, recycling, and transcytosis [2,48,49,50,51,52,53,54]. Herein, we review the expression of FcRn and its control of IgG distribution across tissues (Figure 2).

### 3.1. Endothelium and Hematopoietic Cells

Conditional FcRn deletion experiments have identified endothelial and hematopoietic cells as key mediators of the circulating IgG pool [2]. In monocytes, the quantity of FcRn transcripts can be decreased by a polymorphism in the number of tandem repeats of the FcRn promoter. Monocytes isolated from donors with the deleterious polymorphism exhibited reduced adhesion to immobilized IgG at pH 6.0 and pH 6.5 which, barring affinity-impacting changes in protein structure, implies lower cell surface expression [55]. Site-specific FcRn deletion experiments have further identified macrophages as the major hematopoietic driver of FcRn-mediated recycling and support a hierarchy of FcRn expression: macrophages > monocytes > dendritic cells (DCs) ≥ B cells [56]. In the same set of experiments, depleting liver and splenic macrophages (<35% total macrophage population) in FcRn^−/−^ mice increased IgG half-life by ~50%, suggesting basal FcRn plays a significant role in rescuing IgG from macrophage degradation. Less clear, however, is how FcRn expression is distributed among vascular structures and whether it affects IgG trafficking across tissue locales. It would be interesting to map FcRn expression and trafficking polarity differences, if any, of endothelial cells across broad vascular origins. Considering that a substantial fraction of total IgG resides in the extravascular space (~50%), local variation in endothelial FcRn trafficking may significantly affect IgG tissue distribution [57].

### 3.2. Placenta

Placental FcRn is expressed in placental-fetal endothelial cells, CD68+ macrophages, and syncytiotrophoblasts, where it confers humoral immunity via IgG transfer during the third trimester of pregnancy [58,59,60]. Among the five primary classes of immunoglobulins, only IgG is placentally transferred in any significant quantity. The uptake and transcytosis of maternal IgG are attributed to syncytiotrophoblast cells, while placental-fetal endothelial cells are believed to recycle and protect IgG on their luminal, fetal-facing surfaces [59]. Interestingly, though IgG1-4 can all be detected in fetal circulation, there is an especially high quantity of IgG1. Data support a general transfer hierarchy of IgG subclasses, which is believed to be IgG1 > IgG4 > IgG3 > IgG2, although deviation from this hierarchy in regard to IgG2, IgG3, and IgG4 transport has been reported across study populations [23,60,61,62]. As one might expect, binding between FcRn and IgG is critical to this transfer: a recombinant H435A-bearing IgG1 with undetectable affinity for human FcRn exhibited a ~90% reduction in transfer rate ex vivo [63]. More broadly, placental IgG transfer has important implications for immune programming and drug delivery. The antenatal transfer of ICs, therapeutic antibodies, and vaccine-induced IgG have all been proposed or reported and, depending on the context, can be both protective and deleterious in utero [64,65,66].

One area of open debate is the role of other Fc receptors in placental IgG transfer. In earlier studies, FcRn-mediated IgG transfer was found to be glycosylation agnostic [67]. The discoveries of a) enriched FcγRIIIa-mediated effector function in cord blood IgG, and b) increased galactosylation of cord blood IgG have collectively led some investigators to propose that FcγRIIIa contributes to placental IgG transfer via selective interaction with galactosylated IgG [68,69]. There is significant contention over this point. A recent study argues that these findings can be explained by the well-documented preferential transfer of IgG1, which possesses a greater capacity to engage FcγRs relative to other IgG subclasses [70]. These investigators show that galactosylation of IgG1 and IgG2 does not affect their ability to interact with FcRn and FcγRs. Furthermore, they find neither enhancing nor abrogating IgG1 affinity for FcγRs affects placental transfer in FcRn/FcγR humanized mice, whereas enhancing its affinity for FcRn does. This is corroborated by previous work that found no correlation between Fc glycosylation and preferential placental transfer [71].

### 3.3. Liver and Kidneys

The liver and kidneys are essential regulators of circulatory homeostasis, a trend to which IgG is no exception. Quantitative modeling predicts that the liver and kidneys respectively contain ~14.9% and ~13.7% of total IgG [72]. Another modeling study predicts that while wild-type IgG is usually degraded in the spleen, IgG with reduced FcRn binding is primarily degraded in the liver [73]. In this framework, hepatic FcRn protects IgG from liver-localized degradation. However, it remains unclear whether a particular cellular subset plays an outsized role in maintaining IgG homeostasis, although FcRn is expressed in Kupffer cells, sinusoidal epithelial cells, endothelial cells, and hepatocytes [21]. Some studies hint that the primary subset is unlikely to be hepatocytes, as conditional deletion of FcRn in hepatocytes does not affect circulating IgG levels [74,75]. Beyond preventing intracellular degradation of IgG, FcRn may protect IgG from transcytosis and excretion into the bile, as Fcgrt^−/−^ mice were found to have three-fold higher IgG concentrations in the bile [75]. The same study clarified another important function of hepatic FcRn: the regulation of albumin biodistribution. As hepatocytes are the major synthesizers of serum albumin, loss of FcRn results in intracellular albumin accumulation and increased excretion into the bile.

In the kidney, FcRn expression has been found in proximal tubule cells and podocytes [50]. Investigators posit that FcRn expression here protects albumin and promotes IgG excretion, although it should be noted that IgG can also be reabsorbed in the proximal tubule [76]. As the accumulation of IgG in the glomerular basement membrane (GBM) can trigger inflammation and cause tissue damage, it follows that clearance of IgG from the GBM might serve a protective function: mice deficient in FcRn accumulate IgG in the GBM over time and become sensitized to nephrotoxic nephritis [77]. However, these data are not conclusive, as other studies have found FcRn to play an agonistic role in kidney injury. FcRn-null mice have been found to have shorter IgG half-lives, reduced subepithelial IC formation, and lower overall vulnerability to induced anti-GBM disease [78]. Another study found that when treated with IgG taken from renal transplant recipients that developed transplant glomerulonephritis, podocytes only upregulate pathogenesis markers in the presence of functional FcRn [79]. It is possible that in either of these latter studies, the lack of FcRn reduced circulating IgG concentrations enough to compensate for the absence of FcRn-enhanced renal IgG excretion. The details of this countervailing balance remain unclear. Interestingly, FcRn also seems to be required for proper IL-6 production in podocytes, although the immunological consequences of this are still emerging [80].

### 3.4. Other Epithelium

Beyond the placental, hepatic, and renal epithelia, FcRn also mediates IgG trafficking in the gastrointestinal, retinal, urogenital, and respiratory epithelia. Both the small and large intestines express FcRn across cell populations that include enterocytes, goblet cells, and enteroendocrine cells. Herein, IgG is trafficked bidirectionally—a distinct divergence from the unidirectional trafficking of IgA and IgM [81,82,83,84]. It should be noted that while little IgG can be found in the mucosal secretions of the intestine, IgG transfer into the intestinal lumen potentiates antimicrobial immunity [81]. Inversely, transcytosis of IgG and ICs from the luminal compartment into the lamina propria facilitates immune priming and response [85,86].

Early in life, intestinal FcRn can also promote passive immunity in some species via transcytosis of ingested IgG into neonatal circulation [23,87]. A quantitative investigation found that FcRn expression in the neonatal rat intestine peaks immediately postpartum before rapidly declining, coincident with early lactation and an increase in maternal serum IgG [88]. Bovines, which receive no IgG antenatally, also secrete IgG primarily into colostrum [89]. In contrast, the majority of immunoglobulins in human colostrum immunoglobulins are of the IgA class. When considered alongside the fact that humans express intestinal FcRn throughout life rather than specifically in early life, it seems unlikely that FcRn plays any bespoke role in absorbing free IgG from the neonatal intestinal lumen [21]. Neonatal passive immunity may instead be interchangeably mediated by placental and intestinal FcRn in a species-dependent manner, with IgG secretion into colostrum as a mechanism to compensate for the absence of antenatal transfer. Noteworthily, however, FcRn may yet confer immunity in the human fetal intestine by promoting the uptake of IgG from ingested amniotic fluid [90].

IgG transcytosis has also been found to contribute to vaginal and rectal immunity. One study found that the vaginal washings of mice administered systemic IgG only contained increased IgG levels if the mouse in question expressed functional FcRn. In the same study, systemically administering IgG specific to herpes simplex virus-2 (HSV-2) only protected against intravaginal HSV-2 challenge in mice with functional FcRn [91]. Similarly, broadly neutralizing antibodies (bnAbs) play a role in neutralizing virions at mucosal membranes. FcRn affinity contributes to the disposition of bnAbs in these locations, as engineered bnAbs with increased FcRn affinity exhibited significantly greater trafficking to macaque gut mucosa [92]. These bnAb variants, which persisted in macaque rectums for over 70 days (vs. 28 days), conferred notable protection against intrarectal challenge with simian immunodeficiency virus.

Other locations in which FcRn is expressed include the skin, the respiratory tract, and retinal epithelial cells. In the skin, FcRn is expressed in keratinocytes and immune cells such as histiocytes and dendritic cells [93,94,95]. Although these tissue-local immune cell subsets may utilize FcRn to potentiate adaptive immune responses, as discussed later in this review, a clear link between skin FcRn and IgG disposition has not been elucidated. However, some studies show that FcRn within keratinocytes may facilitate uptake of anti-mitochondrial antibodies, leading to keratinocyte apoptosis and eventual pemphigus vulgaris [96]. In the nasal epithelium, FcRn levels are sufficiently high to deliver transepithelial IgG1, at least in vitro [97]. Whether this finding can be leveraged toward nasal delivery of Fc-bearing therapeutics is an ongoing area of investigation. Similarly, FcRn expressed in the bronchial epithelium of mice, cynomolgus macaques, and humans can mediate absorption of Fc-fusion proteins [31]. These findings raise the possibility of FcRn-mediated reabsorption of IgG and IgG ICs from the respiratory mucosal membranes. In contrast to these therapeutically favorable effects, however, FcRn-mediated transcytosis in the retinal epithelium and retinal vascular endothelium removes intravitreally administered IgG from the eye by directing it toward systemic circulation [46,98,99,100]. Same-side recycling has also been observed in vitro but varies across studies, cell types, and experimental designs [98,99,100]. Overall, FcRn in epithelial tissues exerts incredibly broad effects on IgG distribution that collectively underwrite a plethora of immunological consequences.

### 3.5. Brain

Less than 0.5% of IgG, whether endogenously produced or exogenously administered, is present in the brain at any given time [72,101]. This low brain concentration is thought by some to reflect the exclusionary effects exerted by FcRn expression in the endothelial cells that constitute the blood-brain and blood-cerebrospinal fluid (CSF) barriers [102,103,104,105]. In particular, FcRn in the capillary endothelium has been shown to efflux intracranially administered IgG [104,105]. However, a number of other studies have found a minimal or complete absence of a role for FcRn in IgG transcytosis in this tissue [106,107,108]. Clarifying the function of FcRn in the brain endothelium will require further study. Given that IgG elevation in the CSF is associated with inflammatory diseases, it may be prudent to probe FcRn-mediated brain transport in the context of overabundant serum IgG.

### 3.6. Tumors

Interestingly, there is evidence supporting a role for both increased and decreased FcRn expression in tumor pathophysiology. However, this has been primarily studied in the context of albumin metabolism and albumin-bound cargo delivery [109]. Clinically, FcRn downregulation has been associated with poor prognosis in patients with non-small cell lung cancer, where FcRn protein levels in cancerous tissues can be as low as 25% of protein levels in adjacent non-cancerous tissues [110]. In the same study, immunostaining revealed that few neoplastic cells express FcRn, with most expression instead attributable to macrophages and DCs. These data are consistent with a previous meta-analysis that linked downregulation of *FCGRT*, which codes for the alpha chains of FcRn, to lower overall survival in patients with hepatocellular carcinoma [111]. The mechanistic bases, if any, for tumoral FcRn effects on IgG homeostasis are currently unknown. However, given the well-established importance of macrophages in FcRn-mediated IgG homeostasis, and the fact that macrophages have been found to constitute up to 50% of a tumor’s total mass, it is not beyond supposition that the dysregulation of FcRn in both neoplastic and tumor-associated cells could affect tumoral IgG disposition [112].

### 3.7. Regulation of FcRn Expression

Although FcRn is expressed nearly ubiquitously, its level of expression is dynamically regulated. Polymorphisms in intron 1 of *FCGRT*, which encodes for FcRn, affects FcRn abundance, mAb distribution between central and peripheral compartments, and the efficiency of intravenous Ig (IVIg) therapy [55,113,114]. Methylation-sensitive transcription factors Zbtb7a and Sp1 as well as the microRNAs *hsa-miR-3181* and *hsa-miR-3136-3p* have been found to affect *FCGRT* expression [115,116]. Others have found that FcRn expression responds to immunological stimuli, with detailed investigations showing upregulation by tumor necrosis factor (TNF)-α through NF-κB and downregulation by interferon (IFN)-γ through JAK/STAT-1 [117,118]. These differential expression levels were functionally significant and affected IgG transcytosis by epithelial cell lines. Based on these data, it is conceivable that a mechanism exists to dynamically regulate IgG transcytosis in response to emergent immunological stimuli. However, whether the persistence of circulating IgG is sensitive to immune mediator-modulated FcRn expression remains to be seen.

## 4. FcRn Effects on Immunity

FcRn has emerged as a pleiotropic regulator of extracellular IgG disposition and intracellular potentiation of immunity. Key to this function is the immune complex (IC), which consists of immunoglobulins bound to their cognate antigen (Ag) in multimers of varying sizes. As IgG ICs retain functional Fc domains, they are subject to FcRn-mediated distribution across tissues and intracellular compartments.

At the intracellular level, multimeric ICs preferentially traffic toward degradative lysosomes [119,120]. Some investigators have proposed that the sheer size of multimeric ICs and the rigidity they impose by cross-linking FcRn could play combinatorial roles in preventing its entry into recycling endosomes [119]. Others have found that DCs use lysosome-associated membrane protein 1 (LAMP1) to direct multimeric ICs into lysosomes, where Ags are processed and transferred to MHC molecules [120]. In these studies, no monomeric IgG was detected in lysosomes which, according to the investigators, underlines the ability of FcRn to discriminate between forms of IgG depending on their inflammatory connotations; forms of IgG not associated with inflammation (monomeric) are selectively rescued while forms of IgG associated with inflammation (ICs) are quickly degraded. As will be discussed, the intricate patterns of FcRn-mediated intracellular IC distribution have manifold consequences that include the potentiation of innate immune responses, the enhancement of Ag cross-presentation, and the induction of immunological tolerance (Figure 3).

### 4.1. Innate Immunity 

Neutrophils are the most abundant phagocytic cells of the body. Phagocytosis of IgG-opsonized pathogens is a key mechanism by which they confer immunity and has been shown to rely upon functional FcRn [121]. Disrupting FcRn, whether by α-chain or β2M knockout, reduces neutrophil phagocytosis of opsonized *Streptococcus pneumoniae* by ~50%. Importantly, the change in phagocytic capacity was insensitive to saturating concentrations of IgG, suggesting that FcRn-mediated recycling of IgG–and its presumed effect on opsonization–was not the primary culprit. Within macrophages, the divergent trafficking of monomeric IgG and ICs has been examined in additional detail. Using primary macrophages from FcRn-humanized mice, investigators found that monomeric IgG could bind surface FcγRs with high plasma membrane residency time, resulting in exclusion from endocytosis [122]. This is a departure from (a) the quicker kinetics associated with IgG macropinocytosis and FcRn-mediated recycling, and (b) the trafficking of ICs upon surface FcγR binding, which results in rapid internalization and lysosomal delivery [123]. The authors further postulate that the rapid degradation of ICs precludes them from FcRn exposure and rescue in the macropinosome, ascribing shorter IC half-life to an absence of rescue rather than a pro-catabolic function of FcRn cross-linking. However, the investigators were careful not to rule out the possibility of FcRn interaction in early endocytic vesicles. Regardless, the collaborative capacity of FcRn and FcγRs in macrophages is supported by a recent study wherein FcγRIIa was found to form ternary complexes with FcRn and ICs [124]. In this study, loss of FcRn-IC interactions impaired the ability of FcγRIIa to induce immune responses to IC such as IL-2, IL-6, and TNF-α production [124]. Finally, FcRn may also indirectly affect the maturation of natural killer (NK) cells despite not being expressed. In the bone marrow of FcRn^−/−^ mice, a more significant proportion of NK cells is an immature subtype, although their absolute numbers are not lower [125]. In addition, NK cells isolated from these mice produce much less IFN-γ in response to stimulation with IL-2 or a combination of IL-12, IL-15, and IL-18. The rich biology of FcRn in these contexts suggests that efforts to define a central role in innate immune response cascades may be overly reductionist.

### 4.2. Adaptive Immunity

Antigen-presenting cells (APCs) process antigen into peptides for presentation on MHC class I and II molecules, an indispensable step in the mounting of an adaptive immune response. Late endosomes and lysosomes are key compartments in which Ag loading occurs, being particularly enriched in MHC-II [126]. FcRn contributes toward DC Ag presentation by enhancing the trafficking of multimeric ICs into lysosomes, an effect not observed in IHH-mutated multimeric ICs deficient in FcRn binding [120]. In addition, despite similar levels of APC migration into their lymph nodes, FcRn KO mice exhibit milder CD4^+^ T cell expansion upon stimulation with ICs [120]. These data indicate that FcRn impacts CD4^+^ priming and expansion by controlling intracellular Ag localization and further reinforce its ability to modulate context-dependent immunological responses.

MHC-I Ag loading also depends on FcRn. While FcγRs are conventionally associated with Ag uptake, release, processing, and ultimate cross-presentation to CD8^+^ T cells, this process appears to depend upon the ability of FcRn to capture ICs and modulate their intracellular trafficking following release by FcγRs [127]. Specifically, CD8^−^CD11b^+^ DCs rely on the ability of FcRn to bind and retain ICs in MHC-I-expressing cross-presentation compartments, a spatial localization that simultaneously protects Ag from phagosomal degradation. The MHC-I molecules are then loaded with Ag and shuttled to the plasma membrane for presentation to CD8^+^ T cells. Other detailed studies show that FcRn expression within DCs enhances CD8^+^ T cell activation against cognate Ags and mediates mucosal immunity against colorectal cancer [128]. Strikingly, this effect is marked not only by increased cross-presentation but also by increased secretion of IL-12, a potent CD8^+^ T cell activator. Moreover, enhanced IL-12 secretion upon stimulation with ICs is only possible when FcRn function is intact within the stimulated DCs. Based on these data, it would be interesting to further inquire if FcRn immunopotentiation varies with APC tissue locale, whether it results in differential secretion of cyto- and chemokines, and how these responses lead to differential immune responses.

### 4.3. Immune Tolerance

FcRn supports the establishment of immune tolerance by trafficking ICs across the gut and placental epithelia. The Ags are then processed by APCs and presented to naïve T cells, which become immunosuppressive T_reg_ cells (contingent upon an accompanying milieu of tolerogenic factors). The importance of DCs and Ag-specific immunosuppressive T_reg_ populations has been established and successfully exploited in various preclinical models of autoimmune disease, including hemophilia A and type 1 diabetes [129,130]. Excitingly, a recent study demonstrated that uptake of ICs from breast milk via gut FcRn could generate food allergen-specific T_reg_ cells capable of suppressing allergic responses upon rechallenge, including anaphylaxis [131]. Other studies have found similar roles for intestinal FcRn in the context of allergic airway inflammation and especially highlight the importance of the potent tolerogen transforming growth factor (TGF)-β in milk [132,133,134].

Interestingly, there may also be indirect FcRn-dependent maternofetal transfer of IgE. A study consisting of 144 pregnant women showed a highly significant trend between maternal and cord blood concentrations of IgE/anti-IgE IgG ICs [135]. ICs of this nature strongly bound hFcRn-expressing MDCK cells and were able to be transcytosed bidirectionally. Collectively, these data support an FcRn-dependent mechanism for the maternofetal transfer of IgE in the form of polyclonal ICs. Although it is still unclear how IgE transfer affects tolerogenic immunity in early life, it may warrant reevaluation of the assumption that maternofetal immune functions are conferred primarily by IgG [136].

## 5. FcRn Effects on Monoclonal Antibody Pharmacokinetics

Concurrent with the functional elucidation of FcRn, FcRn-mediated recycling has been enthusiastically exploited to develop mAb therapeutics with reduced dosing requirements and increased potency. Strategies employed in the past, as well as a few more recent advances, are reviewed in this section.

### 5.1. Increased Affinity for FcRn

One straightforward approach to improve mAb persistence is to increase the affinity of FcRn-IgG interactions within the endosome. In rhesus monkeys, IgG2 engineered for modest increases in FcRn binding (3 to 7-fold) at pH 6.0 exhibits nearly double the elimination-phase half-life of wild-type IgG2 [137]. Similar results have also been obtained with IgG1 in hFcRn transgenic mice and cynomolgus monkeys [133]. Over time, several variants such as LS (M428L/N434S), YTE (M252Y/S254T/T256E), and KF (H433K/N434F) have risen to the forefront due to their reproducible ability to extend mAb half-lives, including exceptional cases in which half-life was extended to over 100 days [46,138,139,140,141,142]. These Fc variants, however, also demonstrate increased affinity for FcRn at pH 7.4 and reduced affinity for FcγRs, which potentially limits their effector functions and half-life extension capacity [143,144]. A recent study presents DHS (L309D/Q311H/N434S) Fc variants as a potential solution to these limitations [145]. When introduced to any IgG subclass, DHS mutations significantly improve persistence by increasing FcRn affinity at pH 6.0 and not at pH 7.4. In particular, the IgG1 DHS variant confers over 60% greater in vivo half-life extension over comparable YTE variants. In addition, DHS mutations do not reduce the affinity of IgG for hFcγRs, preserving at least one axis of antibody-mediated effector function. Another interesting approach is the RE (Q438R/S440E) variant, which is particularly suitable for the autoimmune setting as it exhibits improved hFcRn binding without increased rheumatoid factor binding [146].

It follows, then, that improved FcRn affinity at pH 6.0 does not always translate into improved mAb persistence. For example, the bevacizumab T307Q/N434A Fc variant demonstrated a 6-fold higher affinity for murine FcRn at pH 6.0 yet did not exhibit altered pharmacokinetics in mice [147]. Intriguingly, it demonstrated more potent tumor debulking in patient-derived xenograft (PDX) mouse models through unknown mechanisms. In any case, the effort to establish a quantitative in vitro-in vivo relationship between FcRn affinity and pharmacokinetic characteristics is ongoing. One reason that engineering increased FcRn-IgG binding at pH 6.0 may not result in improved half-life is that doing so can also increase binding at pH 7.4 [136,148,149,150]. There are significant negative pharmacokinetic consequences of doing so. For example, one study engineered two anti-TNF-α mAbs with increased FcRn affinity and compared them against wild-type anti-TNF-α mAb in cynomolgus monkeys and mice [148]. In monkeys, where both engineered mAbs had higher FcRn affinity at pH 6.0 than unmodified mAb, exposure, as measured by area under the curve (AUC), was significantly increased. In mice, however, the affinities for murine FcRn at pH 7.4 differed by over tenfold between engineered mAbs, with the stronger binding variant clearing ~3x more quickly. The mechanisms by which clearance is thus enhanced are currently unclear, but impaired exocytosis, proteolysis, endocytosis and degradation, and nonspecific binding have all been speculated to play a role.

In the broader literature, several reports have been published corroborating these findings. One primate study found that while an IgG1 Fc variant with 4-fold increased FcRn affinity at pH 6.0 exhibited increased half-life and slowed clearance, a variant with 80-fold increased FcRn affinity at pH 6.0 did not because it also exhibited substantial FcRn binding at pH 7.4 [143]. Another study found that high FcRn affinity at pH 7.4 could reduce mAb half-life below normal levels [149]. In this scenario, a motavizumab variant with a high affinity for FcRn at pH 7.4 exhibited 6- and 16-fold higher clearance than unmodified motavizumab and its YTE variant, respectively. Subsequent combinatorial efforts to relate the panoply of in vitro mAb characteristics to in vivo pharmacokinetic properties further highlighted the strong relationship between an antibody’s pH 7.4 FcRn affinity and its clearance [150]. Through these aggregate efforts, it has become clear that the relationship between FcRn-IgG affinity at pH 6.0 and half-life extension is contingent upon retention of poor binding at pH 7.4. Ultimately, pH-indiscriminate increases in FcRn binding affinity may undercut realized pharmacokinetic benefit.

Interestingly, engineering mAb-like molecules with additional Fc domains may not increase half-life as one might expect. Fc tandem molecules, which were created by appending additional Fc domains to mAb Fc regions for a total of two to three Fc per molecule, cleared significantly more quickly than unmodified mAb [151]. However, the additional Fc regions did confer IgG2 and IgG3 variants greater affinity for C1q and FcγRs without inducing cytokine release syndrome. The investigators attribute the diminished half-life of these Fc tandem molecules to FcγR binding, which causes a sink effect by promoting endocytosis and degradation. In another interesting study, a group of investigators chose to attach an additional Fc to the light chain of a mAb [152]. This resulted in a 35-fold increase in FcRn affinity at pH 6.0, although pharmacokinetic data were not presented. Notably, the distal nature of the two Fc regions allowed double the typical avidity effect (i.e., two Fc bound to four molecules of FcRn). It will be interesting to examine whether these molecules can persist even longer than other approaches, as well as how they traffic intracellularly relative to multimeric ICs.

### 5.2. Target-Mediated Ag Accumulation and Recycling Antibodies

Soluble, systemically circulating Ags are common targets for therapeutic mAbs. However, soluble Ags may also be subject to FcRn-mediated recycling, as being bound to their cognate mAbs extends their persistence [153,154]. This undesirable effect has motivated the development of mAbs capable of releasing bound Ags intracellularly, thereby diverting them from FcRn-mediated recycling. Borrowing from the pH dependence of interactions between Fc and FcRn, some investigators have engineered pH dependence into Fab and Ag binding such that upon entry into acidic endosomes, a decrease in affinity causes Ag release [155,156,157]. This then allows FcRn-mediated rescue and exocytosis of the mAb while retaining the Ag intracellularly for lysosomal degradation. Antibodies designed in this way can mediate repeated clearance of soluble Ags and are termed “recycling antibodies” (Figure 4). An early study demonstrated the recycling antibodies’ potential to reduce circulating levels of soluble targets such as IL-6R [155]. With transgenic mice expressing hIL-6R, investigators designed a mAb variant with ~22x higher ratio of pH 6.0/pH 7.4 K_D_ relative to tocilizumab that reduced hIL-6R levels six times more quickly. In addition, variations that improved affinity for FcRn at pH 6.0 did not extend the half-life of tocilizumab in cynomolgus monkeys but did so substantially for the recycling variant. It is possible that any pharmacokinetic benefit enhanced FcRn-mediated recycling might have granted tocilizumab was overshadowed by significant Ag-mediated clearance, a persistence-reducing effect. This would have less dramatically impacted the recycling variant that cleared soluble Ag more quickly.

Supporting this point, some have also observed that in these instances of significant target-mediated clearance, the rate of mAb dissociation from Ag (*k_d_*) in acidic environments may be a better predictor of antibody clearance than the relative Ag affinity at acidic and physiologic pH [156]. The potent ability of recycling antibodies to clear soluble Ag can therefore be attributed to their intracellular unbinding of Ag, endosomal binding to FcRn, and subsequent exocytosis, rather than their affinity for Ag alone; moreover, these trafficking pathways have been confirmed by microscopy studies [157]. Of all the recycling antibodies under development, satralizumab became the first recycling antibody to receive commercial approval by the U.S. Food and Drug Administration (FDA) and is indicated for neuromyelitis optica spectrum disorder [158]. Recycling antibodies have since been investigated against other soluble targets including PCSK9, C5, and CXCL10 [156,159,160,161]. In addition, other methods to confer context-dependent Ag binding have been developed by taking advantage of unique physiologic conditions within the endosomal lumen. For example, the relatively low concentration of intraendosomal calcium has motivated the development of mAbs with calcium-dependent Ag binding [162].

### 5.3. Increased Antigen Endocytosis by FcRn Binding

The plasma membrane pool of FcRn plays a limited role in target-mediated endocytosis of endogenous IgG, as binding between the two at pH 7.4 is minimal. Instead, most antibody uptake is via nonspecific fluid-phase pinocytosis. Removing the pH dependency of Fc-FcRn binding by adding affinity at pH 7.4 can dramatically enhance the clearance of soluble Ags when done in conjunction with pH-dependent Ag binding [163]. These “sweeping antibodies” trigger their own endocytosis by binding surface FcRn with Fc regions possessing enhanced pH 7.4 affinity. Upon endocytosis and endosome acidification, the pH-dependent Fab regions release Ag for lysosomal degradation. The antibody is then shuttled back to the cell surface. However, because their Fc regions retain the capacity to bind FcRn at pH 7.4, sweeping antibodies do not dissociate into circulation. Rather, they are retained at the extracellular surface and can be rapidly reuptaken upon sequential Ag binding. Compared to conventional antibodies, equal doses of sweeping antibodies can reduce soluble Ag concentrations by as much as 1000-fold [164]. Unfortunately, and perhaps due to their impaired recycling, sweeping antibodies themselves typically have shorter half-lives [165]. Nevertheless, their superior ability to clear soluble Ags is a considerable pharmacodynamic advantage.

### 5.4. High-Dose COVID-19 Antibodies: Is FcRn Saturated?

Bamlanivimab and REGN-COV2 are IgG1-based antibody therapeutics authorized for early treatment of COVID-19 under an Emergency Use Authorization (EUA). These neutralizing antibodies target the SARS-CoV-2 spike protein and are noteworthy for their high dosing requirements [166,167]. As IVIg is commonly utilized to saturate FcRn and reduce the half-life of its recycled partners, some have inquired as to whether the authorized doses of these antibodies might unintentionally effect similar consequences. In practice, common IVIg doses range from 400 mg/kg and to >1000 mg/kg [168]. Quantitative studies of IVIg administration in immunodeficient patients show a dose-dependent decrease in IgG half-life, albeit somewhat tenuously; patients administered 346 mg/kg IVIg per month saw a ~23% reduction in IgG half-life relative to patients administered 100 mg/kg IVIg per month [169]. As bamlanivimab is approved for use as a single 700 mg IV infusion (10 mg/kg for a 70 kg individual), it is unlikely that it saturates FcRn [166]. A single IV infusion of REGN-COV2, which contains 1200 mg each of casirivimab and imdevimab, still represents only ~34 mg/kg for a 70 kg individual and is likely below the FcRn saturation limit [167]. We therefore consider it unlikely that the authorized doses of these neutralizing antibodies significantly impact the persistence of IgG, mAbs, and other molecules affected by FcRn-mediated recycling.

## 6. FcRn as a Modular Tool and Therapeutic Target

Beyond its exploitation in the rational engineering of mAbs, FcRn-mediated recycling has also been a key component of other therapeutic strategies. The mechanisms of these inspired approaches range from Fc-fusion protein therapeutics to direct FcRn antagonism by anti-FcRn antibodies. An overview of these approaches is provided below and is intended to provide the reader a high-level perspective.

### 6.1. FcRn as a Tool

One modality that clearly stands to benefit from interaction with FcRn is the bispecific antibody (bsAb). These antibodies are engineered to interact with multiple targets simultaneously. Depending on their format and degree of structural modification, they may be subject to a variety of forces that modulate their pharmacokinetic properties. These may include mechanisms shared with mAbs, such as FcRn-mediated recycling and transcytosis, to unique clearance pathways, such as degradation by liver sinusoidal endothelial cells [170]. One group of investigators that sought to increase the systemic clearance of an intravitreally administered bsAb did so by introducing an IHH mutation into its Fc region, ablating affinity for FcRn [171]. In that scenario, a shorter half-life was desirable in order to avoid potential cardiovascular toxicities associated with systemic anti-VEGF treatment. Even without deliberate engineering, other bsAb formats with Fc modifications have been found to exhibit relatively short half-lives, suggesting the presence of recycling deficiencies; conclusive mechanistic evidence, however, remains lacking [172]. Preserving FcRn-mediated recycling may be an important task when attempting to generate bsAbs with clinically favorable persistence.

Many therapeutic proteins have been designed around the understanding that a functional Fc domain, even if synthetically attached, can confer FcRn-mediated recycling. Since the initial approval of the TNFR2-Fc fusion protein etanercept in 1998, over ten additional Fc fusion proteins have been authorized for therapeutic use in the United States [173,174]. The pharmacologically active domains of these proteins are diverse and derive activity from endogenous proteins including vascular endothelial growth factor receptor (VEGFR), glucagon-like peptide-1 (GLP-1), thrombopoietin, and various coagulation factors, among others. Due to FcRn-mediated recycling, Fc-fusion proteins may be amenable to less frequent dosing, an important consideration in the context of injection-limited protein therapeutics. To this point, one recent phase II study reported the pharmacokinetics of HL2351, a fusion protein consisting of two anakinra (hIL-1Rα) components and one Fc domain [175]. Impressively, HL2351 remained in circulation 7-11 times longer than anakinra alone; with comparable pharmacodynamics, HL2351 may be significantly more convenient to administer than anakinra, which requires daily dosing. Though these results are dramatic and reflect the broader potential of Fc fusion to extend therapeutic protein half-lives from mere hours to several days, it is worth noting that these extended half-lives are generally still shorter than those of IgG and mAbs. Given the myriad factors that influence FcRn affinity beyond the Fc region, e.g., Fab-mediated effects, as well as the propensity for target-mediated clearance pathways, perhaps this is unsurprising [33,34,35,36,37,38,174].

Numerous variations on Fc protein fusion, such as Fc binding peptide addition, Fc multimerization, and albumin fusion, have been employed and are reviewed elsewhere [174,176]. The degree to which FcRn affects the pharmacokinetics of these fusion proteins varies with context and can depend on their size and therapeutic target. Two examples that illustrate this point are hexameric Fc-fusion proteins and FcRn binding peptides [177,178,179]. Hexameric Fc generated to dampen autoimmune effector functions could block FcγR-mediated platelet phagocytosis in vivo but did not appear significantly dependent on FcRn affinity for their persistence [178]. Under these circumstances, significant FcγR cross-linking likely contributed to rapid endocytosis and degradation. Conversely, in another set of experiments where a fluorophore was modified to contain various terminal Fc-binding peptides, the extent of in vitro recycling and transcytosis correlated well with peptide affinity for FcRn at pH 6.0 [179]. These data demonstrate the considerable biological complexities associated with FcRn-directed Fc engineering and caution against a one-size-fits-all approach.

Two of the many other modalities that take advantage of FcRn-mediated recycling are engineered antibody domains (eAds) and Fab-dsFv. Engineered antibody domains are antibody-derived therapeutics that possess enhanced tissue penetration due to their smaller size (12–50 kDa). Unfortunately, this is below the glomerular filtration limit and, when compounded with their inability to bind FcRn, causes rapid clearance by renal elimination [177]. One study demonstrated that eAds connected to an FcRn binding motif could persist longer in cynomolgus monkeys despite glomerular filtration, highlighting the potency of FcRn-mediated recycling. There was also a concentration-dependent enhancement of transcytosis in vitro, which raises the possibility of altered eAd disposition. In contrast, Fab-dsFv are Fab domains fused to the Fv domain of an anti-albumin antibody which, in principle, affords an alternative method of attaining FcRn-mediated recycling [180]. The Fab-dsFv had a half-life of 7.9 days in cynomolgus monkeys and 14–17 days in humans after allometric scaling, a significant improvement over the 7–20-h half-life of Fab in humans [180].

### 6.2. FcRn as a Target

FcRn plays an intricate role in autoimmune diseases because it maintains circulating levels of autoreactive IgG while simultaneously promoting the intracellular degradation of ICs. Regarding the prior mechanism, numerous strategies have been employed to inhibit or saturate FcRn-mediated recycling across a broad range of indications; the gamut of FcRn-targeting therapies currently in clinical trials was recently reviewed [181]. One well-established method for saturating FcRn and reducing FcRn-mediated recycling involves intravenous or subcutaneous administration of extraneous IgG–IVIg and SCIg, respectively (Figure 5). However, effective therapy with IVIg or SCIg can require IgG doses above 1.0 g/kg and is subject to worldwide supply shortages [182]. As a result, alternative methods to dampen FcRn activity have been developed and are reviewed below. It is worth keeping in mind, however, that FcRn saturation or inhibition by any means may have broader pharmacokinetic effects than anticipated and could unintentionally reduce the half-life of therapeutic mAbs [183].

Some of the earliest FcRn-blocking reagents developed were “Abdegs”, IgG with Fc domains selectively engineered for higher acidic and physiologic pH FcRn affinity [184]. In mice, Abdegs could dose-dependently inhibit FcRn recycling and reduce circulating IgG ~33–50% within 24 h of intravenous injection. Since these pioneering experiments, additional FcRn-targeting modalities have been developed that collectively represent an appealing alternative to the standard, more broadly immunosuppressive treatments used to treat autoimmune diseases. Of particular relevance is myasthenia gravis, a condition where autoantibodies generated against components of the post-synaptic neuromuscular junction cause muscle weakness and fatigue [185]. The therapeutics currently undergoing clinical testing in this setting include IgG1 mAbs (nipocalimab, RVT-1401), IgG4 mAbs (rozanolixizumab, SYNT001), and IgG1 Fc fragments (efgartigimod) [186,187,188,189]. Early clinical data suggest that all of these can reduce circulating IgG levels to a nadir of ~25–50% at their respective doses, with only mild or reversible effects on albumin.

It is also worth mentioning that autoantibodies have become an important aspect of SARS-CoV-2 infection, particularly in male patients [190,191]. Indeed, autoantibodies generated against type I interferons may impede patients’ ability to control viral load, potentially contributing to longer or more severe infection. However, one group has also detected autoantibodies against a broad range of tissue- and immune-associated targets and found correlations with clinical markers of inflammation and oxygen saturation [191]. Importantly, the persistence and contribution of these autoantibodies to “long COVID” remains unclear; if substantial, it would be interesting to investigate if FcRn blockade applied in the chronic setting could accelerate the clearance of these deleterious antibodies. Of course, any potential upside should be weighed against the risk of compromising antibody-mediated, humoral immunity to SARS-CoV-2 reinfection.

FcRn blockade is also being investigated for other conditions such as pemphigus, warm autoimmune hemolytic anemia, immune thrombocytopenia, and hemolytic disease of the fetus and newborn [181]. Preclinically, FcRn blockade has demonstrated utility in models of epidermolysis bullosa acquisita, arthritis, encephalomyelitis, and thrombocytopenia [192,193,194,195]. Affibody peptides against FcRn have also shown promising results in vivo, but a recent phase I termination may hinder progress in these pursuits [196,197]. As the field evolves, the broad biological activity of FcRn will only be further reflected in the diversity of indications, modalities, and treatment strategies it is leveraged toward.

Indeed, as expansive as the physiologic functions of FcRn are, so too are the potential consequences of its saturation, substrate competition, or modification otherwise. For example, it would be reasonable to suspect that competition for the limited FcRn-mediated recycling pathway might lead to increased IgG catabolism and hypogammaglobulinemia. In the case of therapeutic mAbs that possess high Fc affinity for FcRn, and thus enhanced recycling, some evidence against this phenomenon has been published [198]. Despite the fact that their Fc domains necessarily outcompete endogenous IgG for FcRn binding, FcRn-enhanced mAbs may be administered at merely too low a dose to significantly impact the circulating quantity of IgG. Other consequences at the cellular and physiologic scales have been illustrated by preclinical and clinical studies of SYNT001, an FcRn-targeting IgG4 mAb [188]. Administration of a single dose to hFcRn transgenic mice impeded IFN-γ production by CD8^+^ and CD4^+^ T cells and could reduce circulating IC AUC by roughly two-thirds. Similar trends were reflected in human patients, for whom a single high dose of SYNT001 reduced circulating IC levels by nearly half. In addition, leukocytes isolated from these patients and stimulated with ICs secreted less TNF-α and IL-6 in the presence of FcRn-blocking F(ab)_2_ fragments. The use of F(ab)_2_ fragments precluded nonspecific Fc effects and supported the idea that FcRn-maintained IC pools drive the production of innate inflammatory cytokines. Despite the stark divergence of these physiological-scale FcRn-IC dynamics from intracellular FcRn-IC dynamics, in which FcRn traffics ICs down degradative pathways, these data are supported by recent studies of IC recycling [119,120,199]. Though the actual clinical utility of this strategy remains to be validated, perhaps a degree of cautious optimism is warranted given the limited alternative methods of clearing IC from circulation (e.g., plasmapheresis) [200].

## 7. FcRn in Translational Pharmacology

Extrapolating the pharmacokinetics of antibody and other Fc-bearing therapeutics from studies spanning preclinical, in vitro, and in silico models is a critical practice that necessarily precedes first-in-human studies. Recent advances in each of these domains as they relate to FcRn are reviewed herein.

### 7.1. Preclinical Models

Nonhuman primates, especially cynomolgus monkeys, have been the models of choice to probe FcRn dynamics due to their relatively robust ability to recapitulate human-like pharmacokinetic properties [201,202]. However, high-throughput screens in cynomolgus monkeys are impractical due to financial and ethical constraints. Mice, on the other hand, are significantly higher throughput. Unfortunately, mice also possess a higher affinity FcRn as well as a greater capacity to form clearance-enhancing anti-drug antibodies (ADA). In addition, mice have lower levels of endogenous IgG, which results in a lower level of FcRn binding competition than is present in humans [203]. Collectively, these characteristics undermine the relevance of wild-type mice as models for human antibody pharmacokinetics [46,201].

Several transgenic mouse models have been developed in the absence of high-throughput preclinical models with human-like FcRn dynamics that express human FcRn to varying degrees [93,145,204,205,206]. Tg276 mice lack mouse FcRn but express human FcRn under the control of a ubiquitous tissue promoter. In contrast, Tg32 mice express human FcRn under the control of a human promoter, resulting in a more restricted pattern of expression [206]. While these models are evident improvements, their tissue-level abundance of FcRn still differs somewhat from that of humans [93]. Both are available in homozygous, hemizygous, and severe combined immunodeficiency (SCID) forms. More recent advances include two mouse models deemed “Marlene” and “Scarlett” that express hFcRn and hβ2m under mouse promoters, as well as all human FcγRs [145]. The Scarlett strain also possesses chimeric IgG1 with human Fc and κ light chains. However, its circulating IgG1 levels are retained at relatively low, mouse-like levels, rendering it incapable of reproducing the magnitude of human IgG competition for FcRn. Nevertheless, these models are exciting advancements and warrant further evaluation to capture their added utility.

### 7.2. In Vitro Assays

Establishing a quantitative relationship between in vitro binding assays and in vivo pharmacokinetics is difficult. The methods of characterizing FcRn binding concerning pH dependency have been reviewed thoroughly and include SPR, cell-based systems, immunoassays, and solution-based approaches [207]. One in vitro metric commonly associated with an extended half-life is increased affinity for FcRn at endosomal pH, as this enables more effective rescue from lysosomal degradation [137,138,139,140,141]. However, an increased affinity for FcRn at endosomal pH may be coincident with increased affinity at physiologic pH [141,148,149,150]. Failure to disaggregate these two parameters during drug design can cause even the strongest FcRn-binding therapeutics to be caught in futile loops of recycling, unable to detach from FcRn upon exocytosis; it follows, then, that the k_off_ from FcRn at physiologic pH is also an important predictor of in vivo pharmacokinetics, with slower k_off_ being associated with faster clearance [25,33,34,35,44,208,209]. A physiologic pH affinity threshold of 860 nM has been described, above which half-life is shortened regardless of binding affinity at endosomal pH [149].

Nevertheless, a clear relationship between clinical-stage pharmacokinetics and FcRn binding characteristics remains elusive, with several studies reporting conflicting results [140,210]. Some quantitative studies implicate the endosomal transit rate as a confounding factor in this relationship [211,212]. Specifically, these studies suggest that because the *k_off_* half-lives of mAbs from FcRn are significantly longer than the duration of their endosomal transit times (6–58 min vs. ~7.2 min), *k_on_* must play an outsized role–although *k_off_* is relatively easier to experimentally modify [5,210,213]. This being the case, SPR assays may be used to extract quantitative estimates of IgG *k_on_* and *k_off_* from FcRn. Moreover, factors such as glycosylation, target-mediated drug disposition (TMDD), and FcRn affinity changes upon Ag binding can dramatically impact a therapeutic’s pharmacokinetics [38,40,207]. It is almost certainly the case that favorable in vitro FcRn binding properties alone do not map directly to favorable pharmacokinetics.

Regarding FcRn binding data interpretation, K_D_ measurements derived from SPR analyses can vary highly with experimental design and should be carefully scrutinized. Reported K_D_ values using immobilized FcRn and immobilized IgG1 have respectively ranged between 10–100 nM and 0.2–2.3 μM, with the variance between protocols being attributed to their different capacities to capture avidity-enhanced binding affinity [33,143,209,214,215]. It has been suggested that IgG immobilization assays may better capture monovalent FcRn-IgG binding K_D_, whereas FcRn immobilization assays better capture bivalent FcRn-IgG binding. In any case, consistency when comparing measurements across studies is key. Microscale thermophoresis assays supporting these data have found FcRn K_D_ values on the same order of magnitude, with a K_D_ of 0.9 μM for human IgG1 at pH 5.5 [216]. Biolayer interferometry, combinatorial, and transcytosis assays have also been developed and validated to varying degrees [150,215,217,218,219].

### 7.3. In Silico Models of FcRn Recycling

Compartmental models have long been used to characterize drug disposition. Physiologically-based pharmacokinetic (PBPK) models (and variations thereof) have been especially useful tools with which to evaluate the recycling of Fc-bearing therapeutics. Briefly, PBPK models allow investigators to simulate concentration versus time profiles of drugs within multiple compartments of interest; more often than not, these compartments correspond explicitly with organs and other physiological compartments [220]. Early efforts successfully captured IgG pharmacokinetics and disposition across a broad variety of settings, including in mice subject to IVIg administration, FcRn KO, and tumor grafting, as well as in rats, monkeys, and humans [221,222]. A subsequent catenary PBPK model was developed that contains five sequential endosome compartments, each with progressively lower pH and gradually decreasing affinity between FcRn and IgG [211]. This model predicted a greater contribution of endosomal *k_on_* than *k_off_* to half-life. Another recently published PBPK model could accurately predict the persistence of recycling mAbs as well as LS and YTE antibody variants in humans and Tg32 mice [223].

Under circumstances such as low drug dosing, however, phenomena such as TMDD may impact pharmacokinetics more strongly than FcRn-mediated recycling [207]. This tends to be valid until drug concentrations are high enough to saturate TMDD, beyond which FcRn gradually becomes a more primary determinant. A recent publication was the first to characterize both TMDD and FcRn-mediated recycling within a population PK model of healthy volunteers [224]. The therapeutic of interest was HL2351, which possesses two hIL-1Rα domains (anakinra) fused to an Fc domain and, as a result, persists much longer than anakinra alone. However, this study used an implicit representation of the competition for FcRn binding exerted by endogenous IgG, which may be inadequate when translating for use in patients suffering from autoimmune diseases and potentially elevated circulating IgG.

Contrary to the complexity of PBPK, parsimonious, smaller models with fewer parameters have become popular to evaluate the pharmacokinetics of IgG, mAbs, and Fc-bearing therapeutics. One early study defined peritoneal, central, and endosomal compartments to model the FcRn-mediated disposition of IVIg and its subsequent saturation [225]. IVIg dose escalation up to 2.0 g/kg reduced the percentage of IgG bound to FcRn by up to 83% and increased the fractional catabolic rate 3.7-fold. Another modeling paradigm is the minimal PBPK model, or mPBPK [212,226,227,228,229]. These models apply reductionist principles to maximize the incremental utility of added model complexity. By explicitly defining an endothelial endosome compartment in addition to leaky and tight tissue compartments, one mPBPK model could capture the pharmacokinetics of several mAbs in humans as well as bevacizumab, a bevacizumab Fc variant, a recycling antibody, and a recycling mAb analog lacking pH-dependent Ag binding in hFcRn transgenic mice [228]. This model was further used to evaluate a theoretical recycling antibody with enhanced FcRn affinity and demonstrated that proportional increases in endosomal *k_on_* and *k_off_* from FcRn could result in comparable clearance of circulating target Ag. Interestingly, the same was not true of changes in *k_on_* and *k_off_* of Ag from mAb: increasing the endosomal *k_off_* was a substantially more effective Ag-clearing strategy than decreasing the *k_on_*_._ A similar mPBPK model was built to investigate the effect of endosomal transit time on mAb pharmacokinetics [212]. This study found that endosomal *k_on_* between FcRn and IgG could influence clearance independently of endosomal transit time, whereas the beneficial effects of reducing endosomal *k_off_* were limited by endosomal transit time. In addition, the modeled relationship between *k_off_* and *K_D_* with IgG clearance was inconsistent across 66 mAbs, further emphasizing the importance of *k_on_*. Both of these studies reached conclusions consistent with previous investigations [211].

Another interesting application of mPBPK was recently published in which the potential of disease protein-mediated competition for FcRn protein binding was broached [229]. Specifically, by decreasing concentrations of the M-protein characteristic of multiple myeloma, daratumumab persistence could be improved significantly. These findings introduce an overlooked consequence of certain disease pathophysiologies (e.g., albumin imbalances): the direct interference of FcRn-mediated recycling. Taken *en masse*, these studies highlight the utility of mPBPK modeling during evaluation of nonlinear pharmacokinetics associated with FcRn trafficking.

## 8. Conclusions

FcRn has become an essential therapeutic platform, target, and pharmacokinetic mediator due to its abundant expression across tissues and species. Its manipulation has inspired multiple therapeutics in the autoimmune space, motivated a promising IC clearance strategy, and eased dosing requirements for many indications. Despite these achievements, however, the potential of FcRn is still emerging. Even its most well-studied interaction–binding to IgG Fc–is influenced by myriad factors including but not limited to Ag binding status, IgG subclass, and Fab composition, all of which has yet to be convincingly deconvoluted. As these factors are clarified and a strong relationship between in vitro binding and in vivo pharmacokinetics is established, it will become markedly easier to evaluate and forecast the pharmacokinetic properties of Fc-bearing therapeutics. Until that point is reached, the field will continue to progress with improvements in in vitro assays, preclinical models, and in silico simulations.

## Figures and Tables

**Figure 1 ijms-22-03048-f001:**
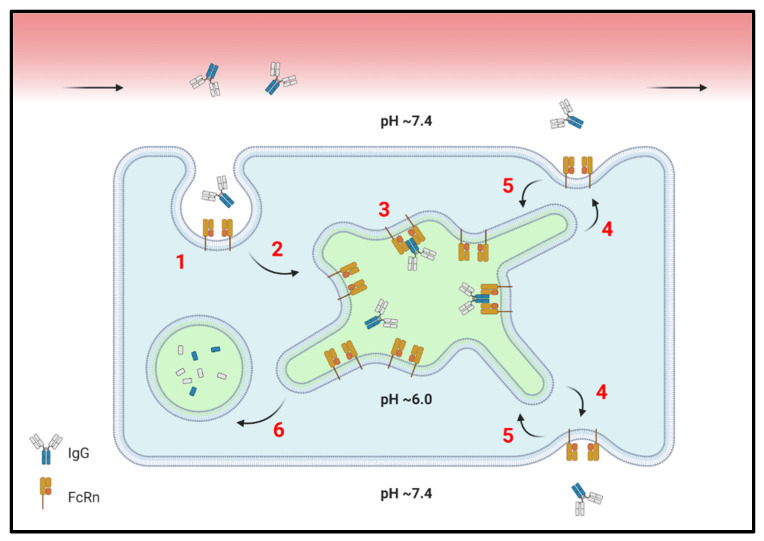
Simplified representation of intracellular IgG salvage by FcRn. (**1**) Uptake of IgG and other circulating factors by macropinocytosis. (**2**) Acidification of the early endocytic vesicle and delivery to the sorting endosome. (**3**) Titration of His residues in the IgG Fc region enables saturable binding to FcRn in a 1:2 complex. Other FcRn substrates, such as albumin, are not depicted for simplicity. (**4**) Recycling endosomes return FcRn-IgG complexes to the cell surface(s), where a shift toward a more neutral pH reduces binding affinity. IgG is then released. (**5**) FcRn is rapidly retrieved from the plasma membrane, resulting in a low surface-resident pool. (**6**) IgG molecules not salvaged by FcRn remain soluble and are delivered by late endosomes (not shown) to lysosomes, where they are degraded.

**Figure 2 ijms-22-03048-f002:**
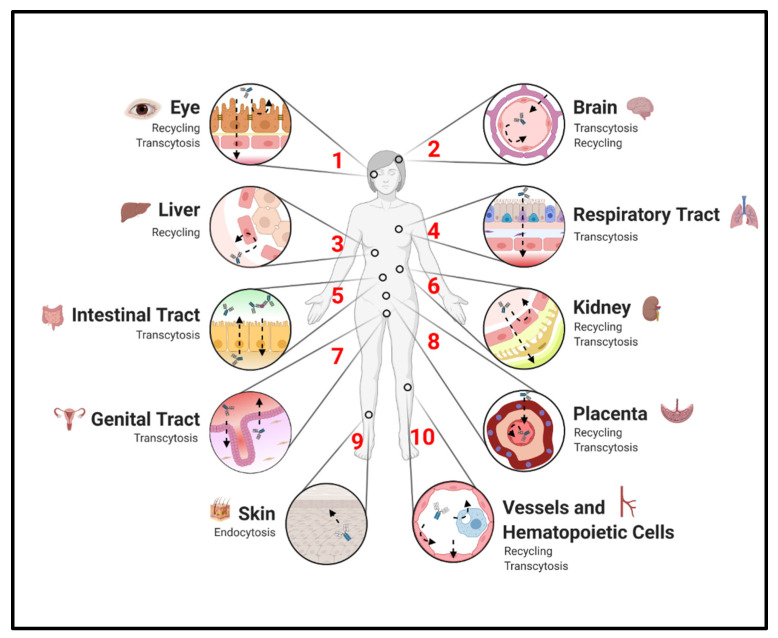
FcRn-mediated effects on IgG disposition across the human body. Generally, endothelial cells mediate recycling, whereas epithelial and parenchymal cells mediate transcytosis. (**1**) In the eye, epithelial and endothelial cells recycle and transcytose intravitreally administered IgG. (**2**) Endothelial cells of the blood-brain barrier and blood-cerebrospinal barrier (not depicted) purportedly recycle IgG and mediate its efflux from the brain parenchyma. (**3**) FcRn in the liver protects IgG from excretion into the bile and is central to albumin synthesis and excretion (not depicted). (**4**) FcRn mediates absorption of IgG and Fc-bearing therapeutics in the lower and upper respiratory tracts. (**5**) Bidirectional IgG transcytosis potentiates antimicrobial immunity in the intestinal lumen and immune priming in the lamina propria. (**6**) The glomerular basement membrane is protected from IgG accumulation by FcRn-mediated recycling as well as transcytosis into the urine. Proximal tubule epithelial cells later reabsorb IgG via FcRn-mediated transcytosis (not depicted). (**7**) FcRn mediates bidirectional transcytosis of IgG in the uterine and vaginal epithelium, contributing to in situ pathogen neutralization and immunity. (**8**) Placental FcRn is expressed in the syncytiotrophoblast, where it facilitates the transfer of IgG from mother to child. Transferred IgG is then maintained by placental-fetal endothelial cells via FcRn-mediated recycling. (**9**) A bespoke trafficking role for FcRn in the skin has not been characterized. However, FcRn expression in keratinocytes can lead to pemphigus vulgaris by promoting the endocytosis of autoreactive IgG. (**10**) The majority of FcRn-mediated recycling is attributable to endothelial and bone-marrow derived cells, especially those of monocytic origins.

**Figure 3 ijms-22-03048-f003:**
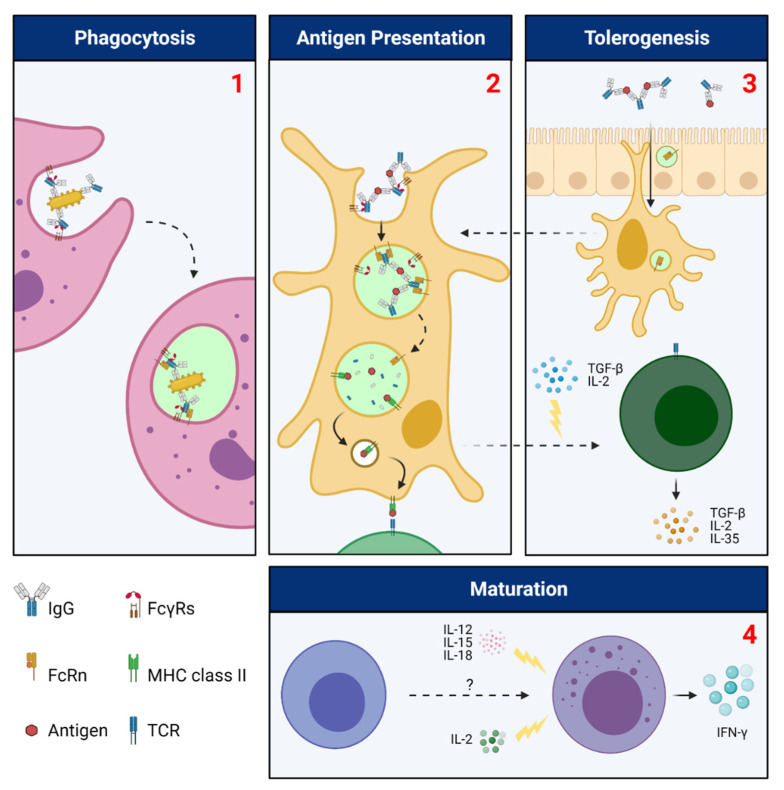
FcRn roles in effector function, antigen presentation, and maturation of immune cells. (**1**) FcRn is critical to phagocytosis. Studies in neutrophils and macrophages suggest opsonized Ag cross-links FcγR on the cell surface, leading to endocytosis and tripartite binding with FcRn. (**2**) Antigen-presenting cells process and present Ag to potentiate adaptive immune responses. Multimeric immune complexes are endocytosed via FcγR cross-linking and relayed to FcRn upon endosome acidification. FcRn directs the multimers toward lysosomes, where they are degraded. Ags, however, are transferred to MHC class II molecules and presented on the cell surface. Some subpopulations capable of cross-presentation on MHC class I molecules also use FcRn to protect Ag from phagosomal degradation (not depicted). (**3**) FcRn-mediated transcytosis of immune complexes in the gut and placenta portends generation of immunosuppressive Treg populations. (**4**) FcRn deficiency impairs NK maturation through unknown mechanisms, suggesting a role in antitumor immunity.

**Figure 4 ijms-22-03048-f004:**
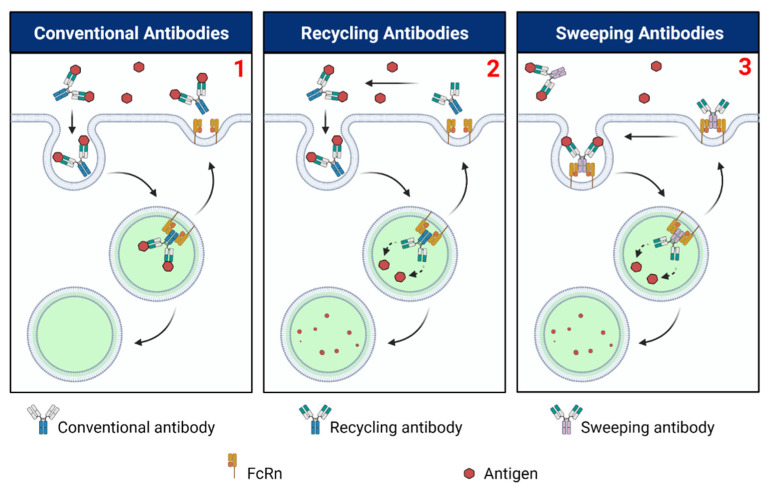
Antigen clearance by recycling and sweeping antibodies. (1) Recycling antibodies enter the cell through nonspecific pinocytosis. By design, recycling antibodies possess pH-dependent antigen affinity. Upon endosome acidification, antigen is released and subsequently degraded in lysosomes, whereas recycling antibodies are rescued by FcRn. The recycling antibody is then exocytosed and free to repeatedly engage additional antigen targets. (2) Sweeping antibodies contain pH-agnostic Fc domains and can bind FcRn at physiologic pH, triggering receptor-mediated endocytosis. Like recycling antibodies, sweeping antibodies possess pH-dependent antigen affinity and follow the same intracellular trafficking patterns. However, sweeping antibodies do not unbind FcRn upon exocytosis, instead lingering at the cell surface. They are subsequently capable of triggering their own endocytosis upon engagement of additional antigen targets.

**Figure 5 ijms-22-03048-f005:**
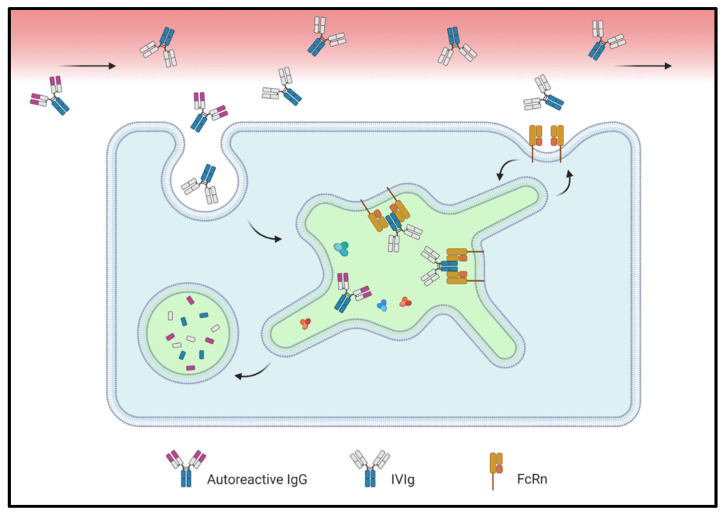
Administration of exogenous IgG saturates FcRn and enhances clearance of endogenous IgG. Competition for FcRn binding reduces the proportion of endogenous IgG protected from intracellular degradation. In autoimmune settings, this may enhance the clearance of pathogenic IgG.

## Data Availability

Not Applicable.

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
