# Peer review of "In Translation: FcRn across the Therapeutic Spectrum"

_ijms, 2021, doi:10.3390/ijms22063048_

Round 1

Reviewer 1 Report

In the manuscript entitled “From Tool to Target: FcRn across the therapeutic spectrum”, the authors examined various aspects of FcRn besides the Fc engineering studies to improve the serum circulating half-life of therapeutic IgG antibodies. In particular, this manuscript well summarized the expression and its roles of FcRn in various human organs and tissues. In addition, the utility of FcRn as a target for potential therapeutic agents to treat various autoimmune diseases.

This paper is a very informative and comprehensive review manuscript on FcRn. It contains well-organized FcRn biology that has been uncovered up to now and a variety of studies to be carried out in the future. Also, it includes antibody engineering strategies for therapeutic purposes by applying such FcRn biological concepts. This reviewer feels this manuscript would be instructive and very helpful for readers who are interested in immunobiology, applied immunology, and antibody therapeutics. I think this manuscript should be accepted after addressing minor comments that I listed below.

  1. Page 2 – 3: A paragraph is italicized. Please correct it.
  2. Line 31-34 on Page 11: The reference [139] is not appropriate for the sentence. This cited article is related to YTE variant, and it does not include contents for LS and KF variants. Therefore, please add another reference or modify the sentence. I would probably add the following reference, Algirdas Grevys 2015 J immuniology
  1. References: some of the references are underlined. Please check the reference style of the journal.

Author Response

Response to reviewer 1

In the manuscript entitled “From Tool to Target: FcRn across the therapeutic spectrum”, the authors examined various aspects of FcRn besides the Fc engineering studies to improve the serum circulating half-life of therapeutic IgG antibodies. In particular, this manuscript well summarized the expression and its roles of FcRn in various human organs and tissues. In addition, the utility of FcRn as a target for potential therapeutic agents to treat various autoimmune diseases.

This paper is a very informative and comprehensive review manuscript on FcRn. It contains well-organized FcRn biology that has been uncovered up to now and a variety of studies to be carried out in the future. Also, it includes antibody engineering strategies for therapeutic purposes by applying such FcRn biological concepts. This reviewer feels this manuscript would be instructive and very helpful for readers who are interested in immunobiology, applied immunology, and antibody therapeutics. I think this manuscript should be accepted after addressing minor comments that I listed below.

Response: Thank you for the positive comments. We have made the following edits to improve the manuscript.

Point 1. Page 2 – 3: A paragraph is italicized. Please correct it.

Response: We have unitalicized the paragraph beginning on Page 2, Line 20.

Point 2. Line 31-34 on Page 11: The reference [139] is not appropriate for the sentence. This cited article is related to YTE variant, and it does not include contents for LS and KF variants. Therefore, please add another reference or modify the sentence. I would probably add the following reference, Algirdas Grevys 2015 J immuniology.

Response: We agree with the reviewer and have added the suggested reference as [145] (Page 13, Line 35).

Point 3: References: some of the references are underlined. Please check the reference style of the journal

Response: Thank you for pointing it out. We have removed the underlining of references in the bibliography

Reviewer 2 Report

The review is well written bringing data together allowing to appreciate all the properties and functions of FcRn. All the facets of this receptor are reviewed based on a high number of citations. I have particularly appreciated the section on the COVID-19 and FcRn. I applaud the authors for the quality of their manuscript I have minor comments and questions for improving the manuscrit.

  1. I would suggest to modify the title. I find the word “tool” inappropriate. When writing “From tool to target”, do you mean something like “the hunted hunter”
  2. The figure 1 can be improved. What are the orange-red three balls? Can you draw FcRn inside the vesicle in formation and adapt the proportion of FcRn molecules on cell surface and inside the cell.
  3. Page 3, concerning the paragraph “Fab contribution to binding”, other data show that mutations located outside the FcRn binding site and not only in the Fab portion, may enhance FcRn binding (in the Fc hinge region: Monnet, C., S. Jorieux, R. Urbain, N. Fournier, K. Bouayadi, C. De Romeuf, C.K. Behrens, A. Fontayne, and P. Mondon. 2015. Selection of IgG Variants with Increased FcRn Binding Using Random and Directed Mutagenesis: Impact on Effector Functions. Front Immunol. 6: 39. doi: 10.3389/fimmu.2015.00039). The kappa or lambda light chains have also been shown to influence binding kinetics to murine FcRn (Gurbaxani, B., L.L. Dela Cruz, K. Chintalacharuvu, and S.L. Morrison. 2006. Analysis of a family of antibodies with different half-lives in mice fails to find a correlation between affinity for FcRn and serum half-life. Mol. Immunol. 43 : 1462–1473). IgG allotypes located in the CH1 and CH3 domain can also modulate FcRn-binding (Ternant D, Arnoult C, Pugnière M, Dhommée C, Drocourt D, Perouzel E, Passot C, Baroukh N, Mulleman D, Tiraby G, Watier H, Paintaud G, Gouilleux-Gruart V. 2016 IgG1 Allotypes Influence the Pharmacokinetics of Therapeutic Monoclonal Antibodies through FcRn Binding. J Immunol. 196:607-613)
  4. The title of the section should be in bold.
  5. In figure 2: Prefer distinguish vessels and hematopoietic cells instead of vasculature.
  6. Page 7: Ref 43 should be changed in 94.
  7. Page 8: In the chapter concerning the “Regulation of FcRn expression” or before in the third section “FcRn expression….”, please mention the polymorphism in the intron 1of FCGRT gene (Sachs UJH, Socher I, Braeunlich CG, Kroll H, Bein G, Santoso S. Avariable number of tandem repeats polymorphism influences the transcriptional activity of the neonatal Fc receptor alpha-chain promoter. Immunology 2006; 119:83–9). Moreover, this polymorphism influences the mAb PK (Passot, C., N. Azzopardi, S. Renault, N. Baroukh, C. Arnoult, M. Ohresser, M. Boisdron-Celle, E. Gamelin, H. Watier, G. Paintaud, and V. Gouilleux-Gruart. 2013. Influence of FCGRT gene polymorphisms on pharmacokinetics of therapeutic antibodies. MAbs 5: 614–619) and IVIG efficacy (Gouilleux-Gruart V, Chapel H, Chevret S, Lucas M, Malphettes M, Fieschi C, Patel S, Boutboul D, Marson MN, Gérard L, Lee M, Watier H, Oksenhendler E; DEFI study group.(2013) Efficiency of immunoglobulin G replacement therapy in common variable immunodeficiency: correlations with clinical phenotype and polymorphism of the neonatal Fc receptor. Clin Exp Immunol. 171: 186-194).
  8. In figure 4: It is not obvious that Ag may also be subject to FcRn-mediated recycling. Please modify the figure that compares recycling Abs and sweeping Abs.
  9. Page 13: It seems that the 153 ref should be changed in 154.
  10. Page 14: In the paragraph “High dose COVID-19….”, why the hypoalbuminemia should modify IgG binding since it is two different binding site?

Author Response

Response to reviewer 2

The review is well written bringing data together allowing to appreciate all the properties and functions of FcRn. All the facets of this receptor are reviewed based on a high number of citations. I have particularly appreciated the section on the COVID-19 and FcRn. I applaud the authors for the quality of their manuscript I have minor comments and questions for improving the manuscript.

Response: we are encouraged by your valuable contribution to the quality of our manuscript. We have made the following edits to improve the manuscript.

Point 1. I would suggest to modify the title. I find the word “tool” inappropriate. When writing “From tool to target”, do you mean something like “the hunted hunter”

Response: We thank the reviewer for their concern and propose a modified title: “In Translation: FcRn across the therapeutic spectrum” (Page 1, Line 2).

Point 2. The figure 1 can be improved. What are the orange-red three balls? Can you draw FcRn inside the vesicle in formation and adapt the proportion of FcRn molecules on cell surface and inside the cell.

Response: We have revised the figure with the recommended changes. The small circular icons were meant to depict other circulating proteins not salvaged by FcRn. They have been removed for clarity (Page 4, Figure 1).

Point 3. Page 3, concerning the paragraph “Fab contribution to binding”, other data show that mutations located outside the FcRn binding site and not only in the Fab portion, may enhance FcRn binding (in the Fc hinge region: Monnet, C., S. Jorieux, R. Urbain, N. Fournier, K. Bouayadi, C. De Romeuf, C.K. Behrens, A. Fontayne, and P. Mondon. 2015. Selection of IgG Variants with Increased FcRn Binding Using Random and Directed Mutagenesis: Impact on Effector Functions. Front Immunol. 6: 39. doi: 10.3389/fimmu.2015.00039). The kappa or lambda light chains have also been shown to influence binding kinetics to murine FcRn (Gurbaxani, B., L.L. Dela Cruz, K. Chintalacharuvu, and S.L. Morrison. 2006. Analysis of a family of antibodies with different half-lives in mice fails to find a correlation between affinity for FcRn and serum half-life. Mol. Immunol. 43 : 1462–1473). IgG allotypes located in the CH1 and CH3 domain can also modulate FcRn-binding (Ternant D, Arnoult C, Pugnière M, Dhommée C, Drocourt D, Perouzel E, Passot C, Baroukh N, Mulleman D, Tiraby G, Watier H, Paintaud G, Gouilleux-Gruart V. 2016 IgG1 Allotypes Influence the Pharmacokinetics of Therapeutic Monoclonal Antibodies through FcRn Binding. J Immunol. 196:607-613)

Response: We thank the reviewer for their interesting addendums and have incorporated their suggestions into the text (Page 5, Line 26; and the paragraph beginning Page 5, Line 43).

Binding modulation by other IgG regions

Beyond the Fab region, other variations distal to the CH2–CH3 juncture also affect binding between IgG and FcRn [41,42]. For example, monoclonal antibody (mAb) therapeutics with G1m17,1 allotypes have been shown to more efficiently bind and be transcytosed by FcRn than G1m3,-1 allotype mAbs. These mutations in the CH1 and CH3 regions of IgG1 and are notable in that variations in either region alone did not affect FcRn binding [41]. Even mutations in the hinge connecting Fc and Fab regions have been found to modulate FcRn binding, although somewhat unpredictably and in a highly heterogenous manner [42]. Taken holistically, it would appear that an IgG molecule’s FcRn binding affinity is highly sensitive to variations along the majority of its length.”

Point 4. The title of the section should be in bold.

Response: We have bolded and unitalicized the title of Section 3, FcRn Expression and its Effects on Homeostatic IgG Distribution (Page 6, Line 15).

Point 5. In figure 2: Prefer distinguish vessels and hematopoietic cells instead of vasculature.

Response: We have adopted the recommended change (Page 7, Figure 2).

Point 6. Page 7: Ref 43 should be changed in 94.

Response: Thank you, that is correct. [46] has been changed to [97] (Page 10, Line 23).

Point 7. Page 8: In the chapter concerning the “Regulation of FcRn expression” or before in the third section “FcRn expression….”, please mention the polymorphism in the intron 1of FCGRT gene (Sachs UJH, Socher I, Braeunlich CG, Kroll H, Bein G, Santoso S. Avariable number of tandem repeats polymorphism influences the transcriptional activity of the neonatal Fc receptor alpha-chain promoter. Immunology 2006; 119:83–9). Moreover, this polymorphism influences the mAb PK (Passot, C., N. Azzopardi, S. Renault, N. Baroukh, C. Arnoult, M. Ohresser, M. Boisdron-Celle, E. Gamelin, H. Watier, G. Paintaud, and V. Gouilleux-Gruart. 2013. Influence of FCGRT gene polymorphisms on pharmacokinetics of therapeutic antibodies. MAbs 5: 614–619) and IVIG efficacy (Gouilleux-Gruart V, Chapel H, Chevret S, Lucas M, Malphettes M, Fieschi C, Patel S, Boutboul D, Marson MN, Gérard L, Lee M, Watier H, Oksenhendler E; DEFI study group.(2013) Efficiency of immunoglobulin G replacement therapy in common variable immunodeficiency: correlations with clinical phenotype and polymorphism of the neonatal Fc receptor. Clin Exp Immunol. 171: 186-194).

Response: The corresponding section has been revised to reference FCGRT polymorphism and its effects on mAbs and IVIg (Page 11, Lines 14 – 16).

“Polymorphisms in intron 1 of FCGRT, which encodes for FcRn, affects FcRn abundance, mAb distribution between central and peripheral compartments, and the efficiency of intravenous Ig (IVIg) therapy [55,114,115].”

Point 8. In figure 4: It is not obvious that Ag may also be subject to FcRn-mediated recycling. Please modify the figure that compares recycling Abs and sweeping Abs.

Response: We have added a panel to the figure illustrating how conventional antibodies might result in recycling of soluble Ag targets (Page 17, Figure 4).

Point 9. Page 13: It seems that the 153 ref should be changed in 154.

Response: We appreciate the reviewer catching this transpositional error. We have swapped the positions of [159] and [160] in the bibliography to correctly map to their in-text citation.

Point 10. Page 14: In the paragraph “High dose COVID-19….”, why the hypoalbuminemia should modify IgG binding since it is two different binding site?

Response: The reviewer raises a fair point. Despite the separate binding sites that allow FcRn to accommodate tripartite binding to IgG and albumin, we feel that we cannot yet preclude potential allosteric effects between the two substrates in vivo. This is primarily due to mixed clinical data wherein nipocalimab, an anti-FcRn mAb ostensibly specific for the IgG binding site of FcRn, has been shown to also deplete serum albumin levels ([187] table S3). However, we understand this is speculation that was not properly elaborated upon in our manuscript. Therefore, we have removed the offending sentence and its accompanying reference [165] (Page 18, Lines 32 – 34).

Furthermore, a significant fraction COVID-19 positive patients are hypoalbu-minemic, which potentially further decreases FcRn receptor occupancy by endogenous substrates [165]”.
